# Two-Stage Multi-Channel Fault Detection and Remaining Useful Life Prediction Model of Internal Gear Pumps Based on Robust-ResNet

**DOI:** 10.3390/s23052395

**Published:** 2023-02-21

**Authors:** Jianbo Zheng, Jian Liao, Yaqin Zhu

**Affiliations:** 1Institute of Vibration and Noise, Naval University of Engineering, Wuhan 430033, China; 2Naval Key Laboratory of Ship Vibration and Noise, Naval University of Engineering, Wuhan 430033, China; 3School of Computer Science and Techonology, Donghua University, Shanghai 201620, China

**Keywords:** deep learning, health status classification, remaining useful life, internal gear pumps

## Abstract

The internal gear pump is simple in structure, small in size and light in weight. It is an important basic component that supports the development of hydraulic system with low noise. However, its working environment is harsh and complex, and there are hidden risks related to reliability and exposure of acoustic characteristics over the long term. In order to meet the requirements of reliability and low noise, it is very necessary to make models with strong theoretical value and practical significant to accurately monitor health and predict the remaininglife of the internal gear pump. This paper proposed a multi-channel internal gear pump health status management model based on Robust-ResNet. Robust-ResNet is an optimized ResNet model based on a step factor h in the Eulerian approach to enhance the robustness of the ResNet model. This model was a two-stage deep learning model that classified the current health status of internal gear pumps, and also predicted the remaining useful life (RUL) of internal gear pumps. The model was tested in an internal gear pump dataset collected by the authors. The model was also proven to be useful in the rolling bearing data from Case Western Reserve University (CWRU). The accuracy results of health status classification model were 99.96% and 99.94% in the two datasets. The accuracy of RUL prediction stage in the self-collected dataset was 99.53%. The results demonstrated that the proposed model achieved the best performance compared to other deep learning models and previous studies. The proposed method was also proven to have high inference speed; it could also achieve real-time monitoring of gear health management. This paper provides an extremely effective deep learning model for internal gear pump health management with great application value.

## 1. Introduction

As the core component of a low noise integrated electro-hydraulic steering device, the internal gear pump has the advantages of simple structure, small size, light weight, good self-priming capability, reliable operation and a low flow pulsation coefficient. However, its harsh operating environment causes accelerated performance degradation during long-term use, and serious acoustic and mechanical faults can occur in the middle and late stages of its life [1]. It is thus clear that internal gear pumps are at risk of reliability hazards and acoustic characteristics exposure during long-term operation [2]. Therefore, it is of great practical significance to diagnose the fault and remaining useful life (RUL) of internal gear pumps.

The main methods of rotating machinery fault diagnosis are physical model-based methods, statistical model-based methods, data-driven artificial intelligence methods, and hybrid methods [3]. With the rise of artificial intelligence technology and the development of sensors and signal processing, data-driven artificial intelligence methods are becoming increasingly popular for rotating machinery fault diagnosis tasks [3,4], which could automatically diagnose the current fault status and RUL of the equipment with little priori knowledge and experience.

Data-driven artificial intelligence methods are divided into machine learning and deep learning methods [3,4]. Machine learning-based fault diagnosis methods usually consist of three steps: data acquisition, feature extraction and pattern recognition. Feature extraction is the focus of these methods and is the most important step in them. For example, Miao et al. [5] used complementary ensemble empirical mode decomposition (CEEMD) and singular value decomposition (SVD) to decompose pressure signal, vibration signal and flow signal to construct feature vectors and built a hydraulic pump fault diagnosis method through a TrAdaBoost migration learning algorithm. Lu et al. [6] decomposed the piston pump outlet pressure signal based on empirical mode decomposition (EMD) and filtered the important intrinsic mode function (IMF) components according to the correlation coefficient criterion, calculating the characteristic energy entropy of each component as the fault feature vector. Yang et al. [7] combined Ensemble empirical mode decomposition (EEMD) signal decomposition and sample entropy to extract rotating machinery fault features. Wang et al. [8] used multiscale alignment entropy as a feature extraction tool for both hydraulic pump and bearing fault diagnosis. Zhou et al. [9] used variational mode decomposition (VMD) to decompose, filter and reconstruct the plunger pump vibration signal, then extracted fine composite multiscale fluctuating dispersion entropy for the processed signal as a feature vector.

Today, with growing computing power and massive amounts of industrial condition data to support then, faults are often reflected in integrated signal feature, which is suitable for deep learning-based methods. Deep learning methods have achieved great success in various fields [10,11], including object detection, segmentation and recognition [12,13,14]. Life prediction and fault diagnosis models based on deep learning can effectively analyse and process these characteristic signals and are already a popular method for predicting the faults and RUL of machinery [15,16].

Typical deep learning methods include deep belief network (DBN) [17], autoencoder (AE) [18], convolutional neural network (CNN) [19,20,21], recurrent neural network (RNN) [22,23] and others. Most have been used in fault diagnosis and RUL prediction. Yuan et al. [24] explored three RNN models, including vanilla RNN, LSTM and GRU, for fault diagnosis and prediction of aero engines. Through extensive experiments they demonstrated that models such as LSTM and GRU performed better than traditional RNN models in RUL tasks. Miao et al. [25] preprocessed sensor signals using the short-time Fourier transform (STFT). Based on a simple spectrum matrix obtained by STFT, an optimized deep learning structure, the large memory storage retrieval (LAMSTAR) neural network, was built to diagnose the bearing faults. Li et al. [26] proposed a deep learning-based remaining useful life (RUL) prediction method to address the sensor malfunction problem. A global feature extraction scheme was adopted to fully exploit information from different sensors, and further, introduced adversarial learning to extract generalized sensor invariant features. Lee et al. [27] used Kalman filter-assisted deep feature learning to evaluate the bi-directional long short-term memory network (Bi-LSTM), which was trained to predict the remaining useful life of an internal gear pump. Xiang et al. [28] proposed an LSTM model based on amplification weights (LSTMP-A) for the remaining life prediction of gears. Compared with conventional LSTMs, LSTMP-A amplifies the input weights and recursive weights of the hidden layers to different degrees through an attention mechanism that depends on the contribution of corresponding data. Wang et al. [29] proposed a scheme to predict the remaining life of a hydraulic pump (internal gear pump) by combining a deep convolutional autoencoder (DCAE) and a bi-directional long and short-term memory (Bi-LSTM) network. The vibration data were characterised using DCAE to construct health indicators and modelled to determine the degradation status of the internal gear pump. Guo et al. [30] proposed a method for predicting the RUL of internal gear pumps based on the forward neural network TrainbrRBFNN. The degradation index was constructed by fusing the various characteristic parameters obtained in the time domain, frequency domain and time-frequency domain. The Trainbr-RBFNN was trained using the simplex fusion index to build an internal gear pump degradation model. The degradation model was used to predict the internal gear pump flow effectively, and the RUL prediction of internal gear pumps was completed.

However, there is still not a solution with integrated, accurate and robust internal gear pump fault detection and RUL prediction. Referring to deep neural networks for image processing, we use the CNN for health state classification and available remaining life prediction tasks. In CNNs, LeNet [31] extracted features by using a combination of successive convolutional and pooling layers. AlexNet [32], proposed by Alex Krizhevsky et al., pioneered the use of deep neural networks to solve image problems. AlexNet has a deeper network structure compared to LeNet; it uses data augmentation, dropout to suppress overfitting and ReLU activation function to reduce gradient disappearance. The AlexNet model achieves better results by constructing a multilayer network but does not provide a direction for deep neural network design. VGG [33] constructs deep convolutional neural networks by using a series of small-sized 3 × 3 convolutional kernels and pooling layers and achieves good results. InceptionNet [34] introduced the Inception block to use different sized convolutional kernels within the same layer of the network, improving model perception, using batch normalisation and alleviating the gradient disappearance problem. However, with the continuous development of deep learning, the number of layers of the model is increasing and the network structure is becoming more and more complex. ResNet proposes residual learning to solve the problem where the training error increases instead of decreasing as the number of layers of the network increases, based on the problems of the above models.

This paper presents a two-stage multi-channel deep learning model for fault detection and RUL prediction based on Robust-ResNet [35]. The use of a step factor h improves the robustness of ResNet and makes it more suitable for tasks in the field of mechanical fault detection. By automatically extracting nonlinear abstract features from massive raw multi-channel data and constructing complex mapping relationships between monitoring data and fault types through a deep network structure without the reliance on complex signal processing techniques and diagnostic experience and enables integrative intelligent fault diagnosis and RUL prediction based on “end-to-end”. The contributions made by the model proposed in this paper, as well as the novelty, are as follows.

(1)A multi-channel two-stage Robust-ResNet-based deep learning method for fault detection and RUL prediction is proposed, which does not require manual feature extraction or complex data processing.(2)The proposed method is based on the Robust-ResNet algorithm with a small step factor h, and the robustness of the proposed method is proved in experiments examining fault detection and RUL prediction in two datasets. Robust-ResNet was used in mechanical fault diagnosis and RUL prediction for the first time. Its superior performance is more in line with the actual scenario of mechanical fault detection. When acquiring data in mechanical work, it is inevitable that noisy data will be obtained; the proposed model is robust, making it more stable and more suitable for generating training data to be used in mechanical fault detection.(3)The proposed method used multi-channel signals in RUL prediction, including pressure pulsation and vibration signals. The fusion of multi-channel signals significantly improved the prediction performance of the method.(4)The model achieves high accuracy in its performance on both tasks, reaching 99.96% accuracy in the fault detection task and 99.53% accuracy in the RUL prediction task in the self-collected internal gear pump dataset. It could also be used in fault detection for rolling bearing, which achieved 99.94% accuracy in the CWRU [36] dataset, which outperformed other state of the art methods. The superior performance of the method on both datasets demonstrates its suitability for multiple application scenarios and that it can be generalised for practical use.

The rest of this paper is arranged as follows. Section 2 describes the details of the method proposed by this work. The experimental setup and results are described and discussed in Section 3. Finally, the conclusion is given in Section 4.

## 2. Materials and Methods

This paper presents a two-stage multi-channel Robust-ResNet-based method for fault detection and RUL prediction of internal gear pumps. This method used the Robust-ResNet network structure to learn the features of internal gear pump data. Robust-ResNet is the optimized model of ResNet, which uses the residual network structure to deepen the deep learning structure. The Robust-ResNet used a small step factor h to improve the performance and robustness of the ResNet. The proposed method contains two stages: in the first stage, the method detects the fault types of the gear pump, including normal type, acoustic fault type and mechanical fault type. In the second stage, the method predicts the remaining useful life of the gear pump, which was divided into eight stages. The proposed method also used multi-channel pressure pulsation and vibration signals to improve the performance of RUL prediction. The framework of the method is shown in Figure 1.

### 2.1. Model Framework

Robust-ResNet [35] is an improved ResNet [37] that is based on an explicit Euler method optimization using a step factor h, which controls the robustness of ResNet in its training and generalization during backward and forward propagation.

#### 2.1.1. ResNet

ResNet is a variant of the deep convolutional neural network (CNN). As the number of layers increases, the more information the deep neural network can acquire and the richer the features that it can learn. However, numerous experiments have shown that increasing the number of layers in a network could result in poorer optimization because of gradient explosion and gradient disappearance. ResNet solved these problems through residual structures for deeper networks. The structure of a residual network block is shown in Figure 2.

Each residual block can be expressed as Equation (1).
(1)$$y_{n}=x_{n}+F\left(x_{n},W_{n}\right);\;\;\;\;x_{n+1}=f\left(y_{n}\right)$$
where $x_{n}$ and $x_{n+1}$ denote the input and output of one residual block, respectively. F is the residual function, and the weighted input $x_{n}$. f is the Relu activation function. Based on the above equation, the learning features from shallow block n to deep block N can be found in Equation (2):(2)$$x_{N}=x_{n}+\overset{N-1}{\underset{i=1}{\sum }}F\left(x_{i},W_{i}\right)$$

#### 2.1.2. Robust-ResNet

Robust-ResNet used the explicit Euler method to describe a ResNet, shown in Equation (3). The step factor h allows for smoother feature transitions, which can prevent information explosion at deep layers while mitigating the detrimental effects of noise in the input features. It could make the feature transitions progressive for each successive block during forward propagation.
(3)$$x_{n+1}≜x_{n}+hg\left(t_{n},x_{n}\right)$$
where $x_{0}≜$
*x*(0), $x_{n}$ is an approximation of $x\left(t_{n}\right)$, h is the step factor, $t_{n}$ denotes the time of transformation along the feature $x_{n}\;$ and $g\left(t_{n},x_{n}\right)\;$ is the partial differential Equation, as in Equation (4).
(4)$$g\left(t_{n},x_{n}\right)≜\partial x_{n}/\partial t_{n}\;\;\;\;\;\;\;$$
given the solution $x\left(t_{n}\right)$ at time $t_{n}$, $g\left(t_{n},x_{n}\right)$ intuitively indicates “in which direction to continue”. At time $t_{n}$ the explicit Eulerian method calculates this direction $g\left(t_{n},x_{n}\right)$ and tracks it over a small time step from $t_{n}$ to $t_{n+1}$. In order to obtain a reasonable approximation, the step factor h in the Eulerian method should be sufficiently small. Its size depends on the partial differential Equation and its given initial input x. The approximation of *x* improves as h decreases.

Then ResNet used the explicit Euler method as described in Equation (5):(5)$$y_{n}≜x_{n}+hF\left(x_{n},W_{n},BN_{n}\right);\;\;\;\;x_{n+1}=f\left(y_{n}\right)$$
where h is the step factor and $BN_{n}$ is the batch normalization.

The batch normalization normalized the inputs on each small batch of training data using Equation (6).
(6)$${\hat{x}}^{\left(k\right)}≜\left(x^{\left(k\right)}-\mu^{\left(k\right)}/\sigma^{\left(k\right)}\right),k=1,2,\ldots d\;\;\;\;\;\;\;\;$$
where x is the input with d dimensions, $x≜{\left(x^{\left(k\right)}\right)}_{k=1,2,\ldots d}^{T}$. $\mu ≜{\left(\mu^{\left(k\right)}≜E\left[x^{\left(k\right)}\right]\right)}_{k=1,2,\ldots d}^{T}$, $\sigma ≜{\left(\sigma^{\left(k\right)}≜\sqrt{Var\left[x^{\left(k\right)}\right]}\right)}_{k=1,2,\ldots d}^{T}$. The batch normalization operation is then performed a mapping transformation on the input using Equation (7).
(7)$$BN\left(x^{\left(k\right)}\right)≜\gamma^{\left(k\right)}{\hat{x}}^{\left(k\right)}+\beta^{\left(k\right)}$$
where $\gamma^{\left(k\right)}$ and $\beta^{\left(k\right)}$ are learned automatically by the network. The learning features of ResNet from shallow block n to deep block N by adding a step factor h can be found in Equation (8):(8)$$x_{N}=x_{n}+h\overset{N-1}{\underset{i=1}{\sum }}F\left(x_{i},W_{i},BN_{i}\right)\;$$
according to the chain rule, denoting the loss function by L, then the gradient of the back propagation can be calculated by Equation (9):(9)$$\frac{\partial L}{\partial x_{n}}=\frac{\partial L}{\partial x_{N}}\frac{\partial x_{N}}{\partial x_{n}}=\frac{\partial L}{\partial x_{N}}\left(I+h\frac{\partial }{\partial x_{n}}\overset{N-1}{\underset{i=n}{\sum }}F\left(x_{i},W_{i},BN_{i}\right)\right)$$
where I denotes the identity matrix. The first term of the loss function, $\frac{\partial L}{\partial x_{N}}$, can propagate the information directly without passing through the weights of any layers, ensuring that no information from $\frac{\partial L}{\partial x_{N}}$ is missing. The second term, $h\frac{\partial L}{\partial x_{N}}\frac{\partial }{\partial x_{n}}\overset{N-1}{\underset{i=l}{\sum }}F\left(x_{i},W_{i},BN_{i}\right)$, is propagated through the weights of the layers, weighting the information of $\frac{\partial L}{\partial x_{N}}$.

Let N=n+1, then $F\left(x_{n},W_{n},BN_{n}\right)≜BN_{n}\left(x_{n}^{\prime }≜W_{n}x_{n}\right)$, where $BN_{n}$ can be calculated in Equation (10).
(10)$$BN_{n}\left(x_{n}^{\prime }\right)≜\gamma \left(x_{n}^{\prime }-\mu \right)/\sigma +\beta$$
let $\sigma_{n}$ denote the smallest component of σ, it can be found that $||\partial F\left(x_{n}\right)/\partial x_{n}||\leq ||\gamma ||\;||W_{n}||/\sigma_{n}$. Finally, a constraint can be found according to Equation (11).
(11)$$||\frac{\partial L}{\partial x_{n}}||\leq ||\frac{\partial L}{\partial x_{n+1}}||\left(1+\frac{h}{\sigma_{n}}||\gamma ||\;||W_{n}||\right)$$
because σ has a tendency to increase, Equation (11) shows that BN has the effect of constraining explosive back-propagation information. However, as the depth network increases, the increase in σ tends to be uncontrollable, so a decreasing step factor h can re-constrain the explosive back propagation of information. A small h could benefit the model.

The loss function for both stages of the model is a cross-entropy loss, and the loss function L is shown in Equation (12).
(12)$$L=-\frac{1}{N}\overset{N}{\underset{i=1}{\sum }}y_{i}\ln{\hat{y}}_{i}$$
where N denotes the number of samples, $y_{i}$ denotes the data label and ${\hat{y}}_{i}$ denotes the predicted value.

In this work, experiments were conducted using h=0.1  and h=1.0 to test the optimisation of small h in Robust-ResNet. The structure of the Robust-ResNet is described in Figure 1. All the residual blocks used 1D convolutional layers in $F\left(x_{n},W_{n}\right)$. The kernel sizes are all 3. The number of feature map is 4, 4, 128, 128, 256, 256, 32, 32 in sequence in each residual block.

#### 2.1.3. Two-Stage Classifiers and Multichannel Signals

In the proposed method, the Robust-ResNet structures were followed by an average pooling layer and a fully connected layer to classify the extracted features. In the first stage of the fault detection task, the first classifier output three types of predicted probabilities corresponding to normal status, acoustic fault status and mechanical fault status. In the second stage, the samples that were predicted to be in acoustic fault status were put into the second Robust-ResNet model for the remaining useful life prediction. We divided these samples into 8 RUL stages according to the time between samples and the final mechanical failure during data collection. Finally, the model output result of the second stage is the probability that the sample remaining life belongs to the eight stages respectively.

In the model for fault detection, the input data are one-channel signals from a pump outlet pressure pulsation sensor. For the second model for RUL prediction, four-channel signals from signals pump inlet pressure pulsation sensor, pump outlet pressure pulsation sensor, pump inlet side acceleration sensor and pump outlet side acceleration sensor were used. We combined signals from each channel into an $N_{c}\times L_{s}$ tensor. $N_{c}$ is the number of channels, which is four in the RUL prediction. $L_{s}$ is the sample length, which is the number of data sample points.

### 2.2. Gear Pump Data Collection

The signals commonly used to monitor and reflect the status and performance of hydraulic pumps include vibration signals [38,39], flow signals [40], pressure signals [41,42], and pressure pulsation signals [43], among which vibration signal analysis is most widely used. According to statistics, vibration signal analysis accounts for about 70% of the existing mechanical fault diagnosis system [44]. However, when the internal gear pump is used as the power source of the ship’s hydraulic system, its working environment is relatively harsh, and the vibration acceleration signal collected through the pump casing not only contains the operating status information of the pump; there is often also environmental noise, vibration of peripheral equipment and other sources of excitation, which will increase the difficulty of real time status monitoring and early fault detection of the pump.

Compared to the vibration signal, the pressure pulsation signal is not easily affected by the surrounding environment and can directly reflect the nonuniform fluctuation of the internal pressure of the pump, so it is more sensitive to the internal status of the pump and early fault [4]. We collected the vibration and pressure pulsation signals collected from an internal gear pump. In the data collection, accelerated life tests were carried out on an accelerated life test rig for an internal gear pump, and the vibration and pressure pulsation signals were obtained for their full life cycle, with each sensor installed in the position shown in Figure 3.

By comparing the level of the outlet pressure pulsation signal, the inlet pressure pulsation signal and the pump casing vibration acceleration signal to characterise the operating status of the pump at each stage of the full life cycle, it can be seen that the outlet pressure pulsation signal of the internal gear pump has the strongest ability to characterise its operating status. We use data from all sensors in the remaining useful life prediction.

The useful life of an internal gear pump is mainly affected by four factors: operating pressure, operating speed, oil temperature and oil contamination level. Based on this, to obtain data on the various fault types of the internal gear pump, an accelerated life test was carried out, divided into an impact test phase and a performance test phase, which consisted of a test on the common operating conditions of the pump and a volumetric efficiency test. To achieve rapid degradation of the performance of internal gear pumps and to ensure that the form of pump fault is consistent with its form of fault in the operating environment, we controlled the four factors in concert. Based on this, combined with the requirements of the machinery industry standard JBT7041-2006 for hydraulic internal gear pump over-speed performance, overload performance and durability performance, we use shock conditions as a means of acceleration, with high speed, high load, high fluid contamination level and high fluid temperature.

The specific experimental parameters for the impact test are shown in Table 1. During the test, 0–20 μm ACFTD fine test dust was added to the oil to increase the cleanliness to NAS11.

Once the pump has completed its initial run, we carry out initial performance tests, including volumetric efficiency tests and common operating conditions tests, to collect operational data on the initial status of the pump, and then conduct pump impact tests. For every 20,000 impacts completed, we complete a performance test on the pump, in a continuous cycle, until the pump fails.

During the accelerated life test, we used various types of sensors to eventually obtain data on the performance status of the pump from the initial stage to the pump fault stage. The final data obtained was in the condition of 1800 rpm & 8 MPa, after 780,000 impacts, plus the initial status, for a total of 40 stages. Each stage included 20,000 impacts. Stages 1–30 were at normal status. Stages 32–39 were in acoustic fault status. Mechanical fault status occurred at stage 40. The eight RUL stages were stages 32–39 of the acoustic fault status. The outlet pressure pulsation signals during its normal and acoustic failure phases are shown in Figure 4.

The gear pump has a sudden change in performance late in the acoustic failure phase. The curve of the volumetric efficiency of the gear pump as a function of the test phase is shown in Figure 5. It can be seen that once the gear pump enters the mechanical failure phase, the volumetric efficiency decreases rapidly and the sudden change in performance only takes up 1/500 to 1/300 of its full life cycle.

## 3. Experiment and Results

### 3.1. Experimental Setup

Our model consisted of a fault detection stage and a remaining useful life prediction stage. In the fault detection stage, the sample length was set as 1024 to sample the continuous 1024 data points. Each sample was sampled continuously and without overlapping. The number of samples are listed in Table 2.

In order to verify the generalization and robustness of the proposed method, we also tested the proposed method in rolling bearing fault diagnosis dataset from CWRU [36] in the fault detection stage. The sample length of CWRU was 512 data points, named CRWU-512. Ten fault types were classified in the datasets, and there were 16,624 samples in the CRWU-512 datasets. The details of the CRWU-512 datasets are listed in Table 2.

In the RUL prediction stage, we tested six proposed models with different sample lengths: 512, 1024, 1536, 1920, 2048 and 4096. The number of training and testing samples are shown in Table 2. Each sample was also sampled continuously and without overlapping.

We tested two proposed models with h=0.1 and h=1.0 in the fault detection stage which were named FD-H0.1 and FD-H1.0, respectively. Six proposed models with h=0.1 were also tested in the RUL prediction stage with different sample lengths. The accuracy of the proposed models was compared with other deep learning models and methods from previous works. The inference time of all the proposed models was also tested to check if the proposed method could meet the needs of real-time monitoring. The parameter settings and data sampling results used in the experiments are shown in Table 3. The hyperparameters used in the model were manually optimised and adjusted during the experiment. Five-fold cross-validation was used in all experiments. The experiments were conducted using Python 3.6 (Python Software Foundation, Wilmington, DE, United States, http://www.python.org accessed on 1 December 2022) and PyTorch 1.8 (Facebook AI Research, New York, NY, USA, https://pytorch.org/ accessed on 1 December 2022) on an Nvidia RTX 2060 GPU (Nvidia Corporation, Santa Clara, CA, USA).

### 3.2. Results of Fault Detection

For the fault detection experiment, we chose data from the pump outlet pressure pulsation sensor. In the fault detection experiment in self-collected gear pump dataset, we compared the proposed models with CNN, CNN + Attention and C/D-FUSA [45] methods. It was found that the proposed methods outperformed other deep learning methods with improved accuracy of 0.16–1.53%, shown in Table 4. The proposed model with a smaller h=0.1 was better than the model with h=1.0, which means that the smaller h improves the performance of ResNet model. In the CWRUdataset, the proposed method still outperformed all the state of the art methods, with improved accuracy of 0.09–1.42%, as shown in Figure 6 and Figure 7, it can be seen that the accuracy curve increases rapidly with the increase of training times, and gradually stabilizes, the mode gradually converges, and the results of cross-validation can also show that the model is both stable and robust, indicating that the proposed method is robust and could be extended to other fault diagnosis tasks, as shown in Table 5. The training and testing accuracy curves of the FD-H0.1 models in the two experiments are shown in Figure 6. It was also found that the accuracies improved steadily in the training phrase.

### 3.3. Results of RUL Prediction

The results of the RUL prediction task are shown in Table 6. Accuracy curves of the six proposed models are shown in Figure 7. In this experiment, we compared the proposed model in datasets with different sample lengths. Our results showed that the proposed model performed best when the sample length was 512, where it had an accuracy of 99.53%. We also compared the performance of different methods and different numbers of channels when the sample length was 1024. In different methods, the proposed model outperformed other methods significantly, with improved accuracy of 6.2–15.49% for multi-channel signals. The accuracy of the proposed method was better than the other methods, showing improvement of 0.7–23.93%. The improvement means that the feature extraction ability of the proposed method was much better than the other deep learning methods.

In addition, in all the methods, the performance of the model based on the multi-channel signal far outperformed that based on the single channel signal. This may indicate that multi-channel signals have more effective information, which were easily discovered and integrated using deep learning methods.

In the analysis of the Robust-ResNet results in the experimental data, it was found that when the sample length was 512, the number of samples in a single class was higher, so the model training accuracy was high. However, when the sample length increased to 1024, the number of samples decreased, and the accuracy also decreased. However, when the sample length was gradually increased to 4096, more feature information was contained in a single sample, and the features of the internal gear pump data could be learned more effectively using the proposed model. When comparing the results of single- and multichannel signals, we found that the multi-channel data contained more features and were better suited to our model for learning.

The experimental results showed that the proposed model had good performance in the small sample dataset and even better performance in the large sample dataset, which was in line with the practical application scenario.

### 3.4. Model Inference Time

Although the proposed Robust-ResNet-based model outperformed other models on the two tasks, the proposed model was also more complex in structure than some of other deep learning methods. Therefore, we also tested the inference time of the model in order to verify whether it could be used in real time monitor applications. To offer additional information, we compared the sampling time and inference time of individual samples in each model. The inference speed of the models was tested in a CPU environment by reading the trained models. We used a single-core 2.5 GHz CPU in this experiment. The sampling frequency of the self-collected dataset was 65,536 samples per second. The results for inference times are shown in Table 7. From the experimental results, it is clear that, for different tasks and different sampling frequencies, the inference speed of our model was less than the sampling time, proving the real time inference ability of our model.

### 3.5. Discussion

CNN is mainly used to train the gear pump data, to realize the gear pump health status classification and for RUL tasks. In convolutional neural networks such as AlexNet, VGG and InceptionNet, the more layers there are, the richer features of different levels can be extracted. The deeper the network is, the more abstract the extracted features are and the more semantic information they have. However, as the number of network layers increases, the model structure gradually becomes more complex, and the network is faced with the problem of degradation, leading to increased model error. Based on this problem, ResNet proposed residual learning, which does not directly fit the output value but learns an identity mapping. However, in the actual working environment of gear pump health status classification, the acquisition of data will inevitably obtain noise data, which will cause adverse effects on the model and affect the final model performance. Such errors may cause significant losses in the actual environment. Therefore, based on ResNet, step factor h is used in this paper to improve the robustness of ResNet and reduce the impact of noise data on the model. This is more in line with the actual application situation. In the health status classification and RUL tasks, according to the accuracy curve, it is obvious that the proposed method can converge quickly to reach a stable state, and has very high accuracy on both data sets, which also proves that our method can be extended to different application scenarios and is not limited to the health status management of gear pumps.

## 4. Conclusions

In this paper, we proposed a two-stage multi-channel Robust-ResNet-based gear pump health monitoring method. The proposed method could deal with two-stage tasks, including fault detection and remaining useful life prediction. The core of the proposed method is a Robust-ResNet model with the enhancement of a small step factor. This method optimized the traditional ResNet model by using the explicit Euler method and enhanced its robustness and performance. The proposed method could also integrate multi-channel pressure pulsation signals and vibration signals to extract more effective features and improved the performance of detection and prediction. The proposed model outperformed convolutional neural networks, the deep learning models with attention, and other state of the art models in the two datasets with different sample lengths. At the same time, the proposed model also had extremely fast inference speed, which could achieve real time monitoring. This work provided a feasible and advanced solution for the integrated gear pump health management and could also be extended to other mechanical applications.

In the task of health status classification, the model is compared with the network structure of CNN, CNN + Attention, etc. and it can be seen that this method is superior to other methods, which verifies the superiority of this method in the task of gear pump health status management. Compared with other models, it can better avoid the influence caused by noisy data. The accuracy of this model for the multi-channel signal is improved by 6.2~15.49%, which is significantly better than that of other methods, and the accuracy of this method is improved by 0.7~23.93%. This improvement means that the feature extraction capability of this method is significantly superior to other deep learning methods. According to the experimental results, the proposed method can not only reduce the influence of noise data but also extract multi-channel signals with high feature extraction ability. These two characteristics make the method not only suitable for the gear pump data health state management but can also be extended to other application scenarios.

## Figures and Tables

**Figure 1 sensors-23-02395-f001:**
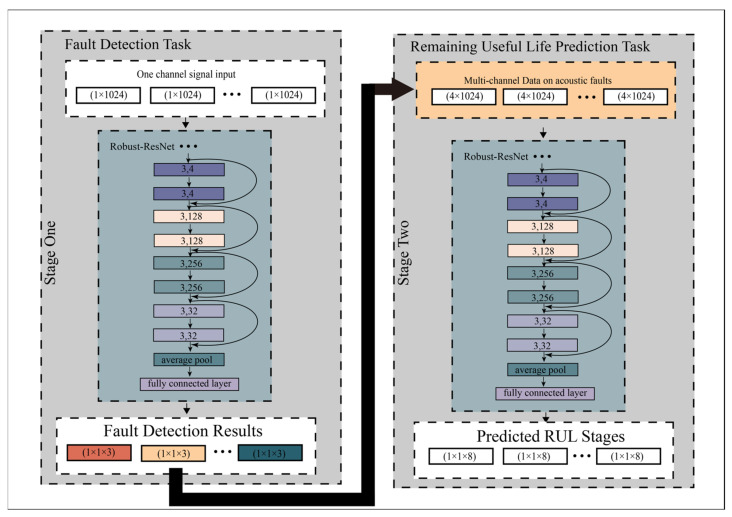
Methodological framework diagram of the proposed method. Our method is mainly based on Robust-ResNet implementation for two-stage tasks: fault detection and remaining useful life prediction. Robust-ResNet achieves robustness of the ResNet network using a small step factor h. The first-stage task is the fault detection using one-channel signals of 1 × 1024 collected from pump outlet pressure pulsation sensor. The samples with acoustic faults are then used in the second stage: remaining useful life prediction. Four-channel signals were used in the RUL prediction, including signals from the pump inlet pressure pulsation sensor, pump outlet pressure pulsation sensor, pump inlet side acceleration sensor and pump outlet side acceleration sensor. The size of the input tensor is 4 × 1024. For the RUL prediction stage in the figure, a sample length of 1024 is used as an example. Different sample lengths are compared in the actual experiment.

**Figure 2 sensors-23-02395-f002:**
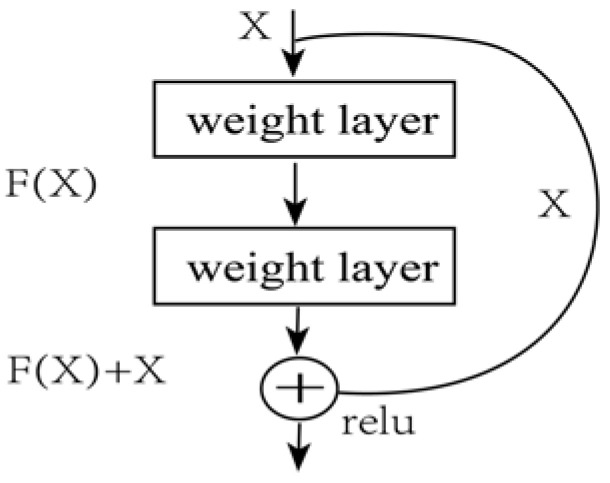
Residual network block structure.

**Figure 3 sensors-23-02395-f003:**
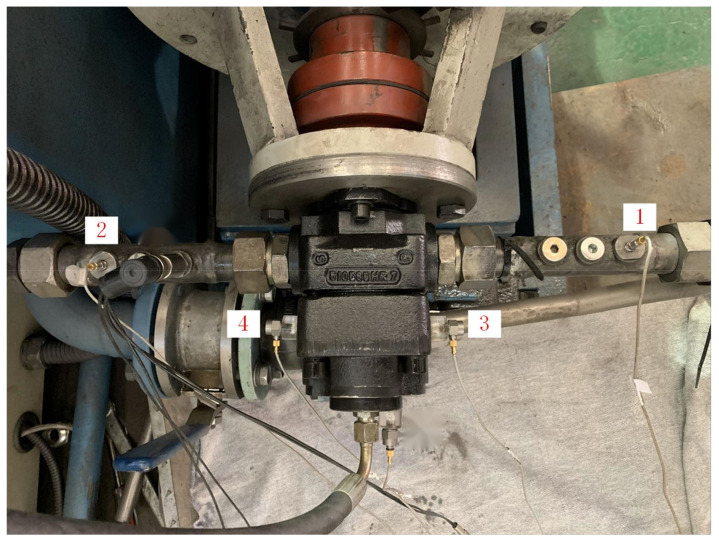
Sensor mounting positions. Four sensors were installed to collect signals: 1. pump inlet pressure pulsation sensor; 2. pump outlet pressure pulsation sensor; 3. pump inlet side acceleration sensor; 4. pump outlet side acceleration sensor.

**Figure 4 sensors-23-02395-f004:**
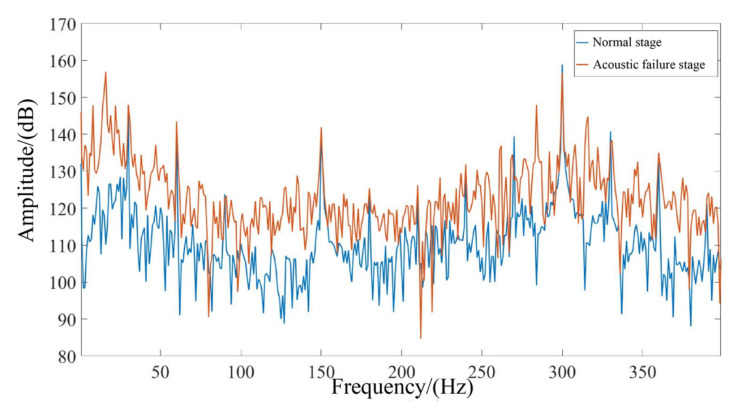
Comparison of frequency domain waveforms in different life stages of internal gear pumps.

**Figure 5 sensors-23-02395-f005:**
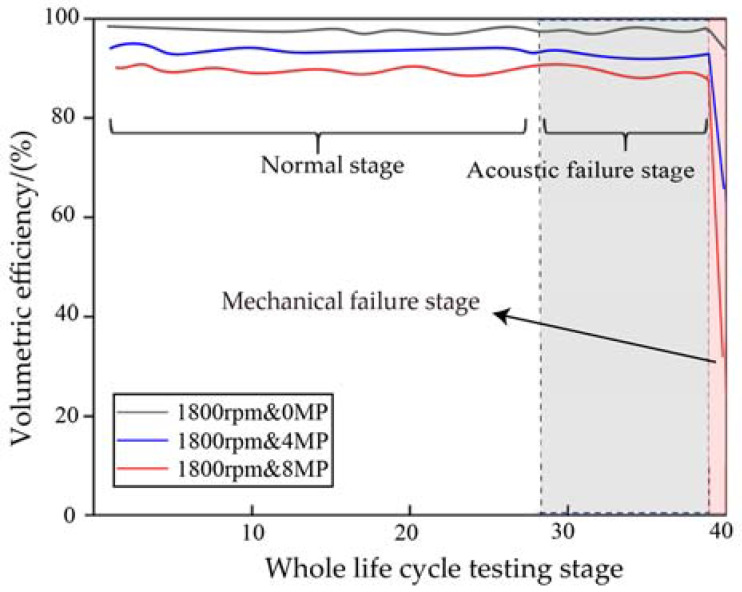
Full life variation curve of the volumetric efficiency of internal gear pumps.

**Figure 6 sensors-23-02395-f006:**
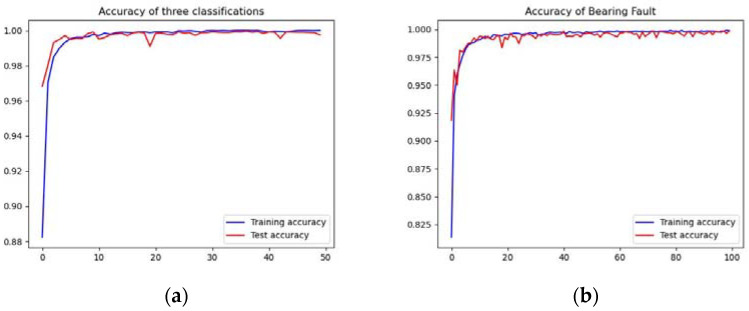
Accuracy curves of the proposed methods on different datasets, where (**a**) is the accuracy graph of the proposed method with h=0.1 for fault detection on the self-collected gear pump dataset and (**b**) is the accuracy curves of the proposed method with h=0.1 on the CWRUdataset.

**Figure 7 sensors-23-02395-f007:**
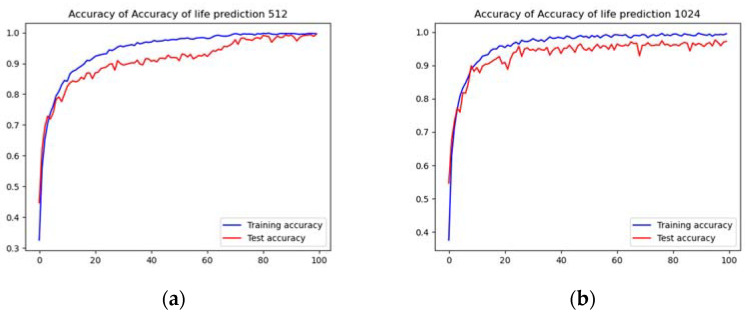
Accuracy curves of the proposed model with different sampling length: (**a**) Sampling length = 512; (**b**) Sampling length = 1024; (**c**) Sampling length = 1536; (**d**) Sampling length = 1920; (**e**) Sampling length = 2048; (**f**) Sampling length = 4096.

**Table 1 sensors-23-02395-t001:** Parameter setting for the impact test phase.

Parameters	Settings
System oil temperature	80 °C
Test speed	Maximum pump speed
Impact pressure	Pump 125% of rated pressure
Impact frequency	40 times/min
Single-step impact (loading time)	1.0 s
Single-step impact (unloading time)	0.5 s

**Table 2 sensors-23-02395-t002:** Details of the CRWU-512 in the fault detection experiment.

Fault Location	Diameter	Number of Samples	Number of Samples in CWRU-512
Normal	Normal	2,182,450	4261
Ball	0.007	487,093	950
	0.014	488,109	951
	0.021	487,964	951
Inner Raceway	0.007	488,309	952
	0.014	487,239	948
	0.021	487,529	950
Outer Raceway	0.007	1,465,051	2855
	0.014	487,819	950
	0.021	1,465,487	2856

**Table 3 sensors-23-02395-t003:** Experimental parameter settings and number of samples in the gear pump dataset.

Model	Learning Rate	Epoch	Optimizer	Loss Function	Number of Training Samples	Number of Testing Samples
FD-H0.1	0.03	50	SGD	CrossEntropyLoss	30,702	7680
FD-H1.0						
RUL-512	0.001	100	Adam		12,288	3072
RUL-1024					6114	1536
RUL-1536					4096	1024
RUL-1920	0.003				3276	820
RUL-2048					3072	768
RUL-4096					1536	384

**Table 4 sensors-23-02395-t004:** Comparison of experimental results for internal gear pumps.

Dataset	Model	Accuracy (%)
Self-collected gear pump dataset	CNN	98.43
	CNN + Attention	99.64
	C/D-FUSA [45]	99.80
	FD-H1.0	99.81
	FD-H0.1	99.96

**Table 5 sensors-23-02395-t005:** Comparison of experimental results for rolling bearings.

Dataset	Model	Accuracy (%)
CWRU	C/D-FUSA [45]	99.85
	Lei et al. (2016) [46]	99.66
	Wang et al. (2022) [47]	99.15
	Yan et al. (2022) [48]	98.52
	FD-H1.0	99.90
	FD-H0.1	99.94

**Table 6 sensors-23-02395-t006:** Accuracy of remaining useful life prediction.

Method	Channel	Model	Accuracy (%)
Proposed model	Multichannel	RUL-512	99.53
		RUL-1536	97.67
		RUL-1920	98.66
		RUL-2048	98.83
		RUL-4096	99.23
		RUL-1024	97.23
	Single-channel	RUL-1024	78.34
CNN	Multi-channel	RUL-1024	81.74
	Single-channel	RUL-1024	52.41
CNN + Attention	Multi-channel	RUL-1024	89.25
	Single-channel	RUL-1024	70.96
C/D-FUSA [37]	Multi-channel	RUL-1024	91.03
	Single-channel	RUL-1024	77.64

**Table 7 sensors-23-02395-t007:** Sampling time and model inference time for different tasks and different sample lengths of data.

Dataset	Sampling Time (s)	Inference Time (s)
FD-H0.1	0.0156	0.0021
RUL-512	0.0078	0.0058
RUL-1024	0.0156	0.0006
RUL-1536	0.0234	0.0006
RUL-1920	0.0293	0.0059
RUL-2048	0.0313	0.0058
RUL-4096	0.0625	0.0059

## Data Availability

Data available on request due to restrictions e.g., privacy or ethical. The data presented in this study are available on request from the corresponding author. The data are not publicly available due to [Experimental privacy].

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
