# Peer review of "Two-Stage Multi-Channel Fault Detection and Remaining Useful Life Prediction Model of Internal Gear Pumps Based on Robust-ResNet"

_sensors, 2023, doi:10.3390/s23052395_

Round 1
Reviewer 1 Report
This paper proposes Two-stage Multi-channel Fault Detection and Remaining Useful Life Prediction Model of Internal Gear Pumps Based on Robust-ResNet. In general, this paper is well presented. The following issues can be further considered.
1. More background and motivation of this study can be added, in case the readers are not very familiar with the topic.
2. The descriptions of the well known knowledge can be properly reduced.
3. Why introducing the multi channel based PHM method for the problem? What is the major benefits compared with traditional methods?
4. Some related works on this topic should be reviewed, such as "Deep learning based approach for bearing fault diagnosis", "Remaining Useful Life Prediction with Partial Sensor Malfunctions Using Deep Adversarial Networks", etc.
5. A couple of ablation studies should be added to evaluate the effects of the key parameters of the proposed method on the performance.
Author Response
Response to reviewer1
- More background and motivation of this study can be added, in case the readers are not very familiar with the topic.
Response: Thank you very much for your suggestion. We compare existing convolutional network algorithms on line 111, page 3, and explore the research context in further depth. We present the advantages and disadvantages of convolutional networks such as LeNet, AlexNet, GoogleNet, and VGG, and describe why we chose Robust-ResNet for the task of gear pump fault detection.
Referring to deep neural networks for image processing, we use the CNN for health state classification and available remaining life prediction tasks. In CNNs, LeNet[31] extracts features by using a combination of successive convolutional and pooling layers. AlexNet[32] proposed by Alex Krizhevsky et al. pioneered the use of deep neural networks to solve image problems, AlexNet has a deeper network structure compared to LeNet, using data augmentation, dropout to suppress overfitting, and ReLU activation function to reduce gradient disappearance. The AlexNet model achieves better results by constructing a multilayer network, but does not provide a direction for deep neural network design. VGG[33] constructs deep convolutional neural networks by using a series of small-sized 3x3 convolutional kernels and pooling layers, and achieves good results. InceptionNet[34] introduced the Inception block to use different sized convolutional kernels within the same layer of the network, improving model perception, using batch normalisation and alleviating the gradient disappearance problem. However, with the continuous development of deep learning, the number of layers of the model is increasing and the network structure is becoming more and more complex. ResNet proposes residual learning to solve the problem that the training error increases instead of decreasing as the number of layers of the network increases, based on the problems of the above models.
- The descriptions of the well known knowledge can be properly reduced.
Response: Thank you very much for your suggestion. We have removed some of the known knowledge from the manuscript.
The deleted parts are as follows:
In line 44, page 1:
The physical model-based approach describes the degradation mechanism of mechanical equipment by establishing a mathematical model, the parameters of which are mainly related to materials, working environment, etc. Statistical model-based methods, mainly by establishing a statistical model based on empirical knowledge, according to the statistical results, the remaining life prediction results in the form of the probability density function, the common statistical models include the Wiener process model, Markov model, AR model, inverse Gaussian process model, etc. However, due to the complex working environment of internal gear pumps in the submarine environment, it is difficult for model-based methods as well as hybrid methods to locate internal gear pump faults quickly and accurately.
In line 288, page 7:
The internal gear pump is a displacement pump, the flow pulsation due to the output flow discontinuity and the impedance characteristics of the pipeline will be directly reflected in the system pressure pulsation.
- Why introducing the multi channel based PHM method for the problem? What is the major benefits compared with traditional methods?
Response: Thank you very much for your suggestion. The data of a single channel represents the data obtained from a pressure sensor, while the multi-channel data includes the data obtained from four sensors. Compared with the data of a single channel, the multi-channel data contains more characteristic information, which also conforms to the advantages of our model. Our model can extract more characteristic information than other algorithms. Multi-channel data information is more beneficial to gear pump fault detection.
- Some related works on this topic should be reviewed, such as "Deep learning based approach for bearing fault diagnosis", "Remaining Useful Life Prediction with Partial Sensor Malfunctions Using Deep Adversarial Networks", etc.
Response: Thank you very much for your suggestion. We have added the relevant literature work in line 82, page 2 of the manuscript.
Miao et al [25] preprocessed sensor signals using short-time Fourier transform (STFT). Based on a simple spectrum matrix obtained by STFT, an optimized deep learning structure, Large Memory Storage Retrieval (LAMSTAR) neural network, is built to diagnose the bearing faults. Li et al [26] proposed a deep learning-based remaining useful life (RUL) prediction method to address the sensor malfunction problem. A global feature extraction scheme is adopted to fully exploit information of different sensors and further introduced adversarial learning to extract generalized sensor invariant features.
- A couple of ablation studies should be added to evaluate the effects of the key parameters of the proposed method on the performance.
Response: Thank you very much for your suggestion. Our experimental section has a related ablation experiment in which we compare the effect of single and multi-channel data on model training, as well as setting different step factor h and experimentally verifying the effect of h on model performance. We also compare existing CNN and CNN+Attention algorithms for single channel data and multi-channel data, and finally validate the advancement and superiority of our algorithm.
And in line 380, page 11:
It could be found that the proposed methods outperformed other deep learning methods with improved accuracy of 0.16%-1.53%, shown in Table 4. Proposed model with smaller was better than the model with , which means that the smaller improves the performance of ResNet model.
In line 402, page 12:
In this experiment, we compared the proposed model in datasets with different sample lengths. It could be found that the proposed model performed best when the sample length was 512 with accuracy of 99.53%. We also compared the performance of different methods and different numbers of channels while sample length was 1024. It could be found that for different methods, the proposed model outperformed other methods significantly, with improved accuracy of 6.2%-15.49% for multi-channel signals. The accuracy of the proposed method was better then the other methods with improvement of 0.7%-23.93%. The improvment means that the feature extraction ability of the proposed method was much better than the other deep learning methods.
In addition, in all the methods, the performance of the model based on the multi-channel signal far outperformed that based on the single channel signal. This may indicates that multi-channel signals had more effective information, which were easy to be discovered and integrated by deep learning methods.

Reviewer 2 Report
In this paper, the authors, proposed a multi-channel internal gear pump health status management model based on Robust-ResNet. Robust-ResNet is an optimized ResNet model based on a step factor ℎ in the Eulerian approach to enhance the robustness of the ResNet model. The following comments must be carefully revised to improve the quality of the paper.
1. Page 4-line 144. The reason why ResNet in Fig. 1 is adopted as the backbone model needs to be further explained. Since its advantage lies in its particularly deep structure, is this necessary for the task of this paper? Whether the four residual blocks can play a performance advantage should be explained, because many other structures like AlexNet and Inception Net can be replaced.
3. Page 7-line 221. How the convergence of the model is guaranteed in two stages should be discussed.
4. Page 10-line 311. The training hyper-parameters (e.g. batch size, etc) of the deep learning model are ignored. The division ratio of the training set and test set and the number of samples should also be reported.
5. Page 14-line 401. Since the author reported the reasoning speed, the experimental platform should be given.
6. Page 12-line 355. The author gives the optimization curve, but should supplement the loss function of the model.
7. Page 1-line 31. Deep learning has demonstrated superior performance in many fields, including object detection, segmentation, and recognition. The following related work of deep learning for generalization and applications must be cited in Introduction, including “Improvement of generalization ability of deep CNN via implicit regularization in two-stage training process,” IEEE Access, vol. 6, pp. 15844-15869, 2018. “Compound figure separation of biomedical images with side loss.” Deep Generative Models, and Data Augmentation, Labelling, and Imperfections. Springer, Cham, 2021. 173-183. VoxelEmbed: 3D instance segmentation and tracking with voxel embedding based deep learning. International Workshop on Machine Learning in Medical Imaging. Springer, Cham, 2021: 437-446. “Pseudo RGB-D Face Recognition,” in IEEE Sensors Journal, vol. 22, no. 22, pp. 21780-21794, 15 Nov.15, 2022.

Author Response
RESPONSES TO REVIEWER1
- Page 4-line 144. The reason why ResNet in Fig. 1 is adopted as the backbone model needs to be further explained. Since its advantage lies in its particularly deep structure, is this necessary for the task of this paper? Whether the four residual blocks can play a performance advantage should be explained, because many other structures like AlexNet and Inception Net can be replaced.
Response: Thank you very much for your suggestion. A comparison of the various network architectures has been made in the manuscript on line 110, page 3, and the reasons for the final choice of Robust-ResNet are given. Firstly, ResNet solves the problem of increasing the number of layers in a deep neural network but increasing the error instead of decreasing it. Robust-ResNet improves the robustness of the network by using a step factor on top of it, which is more suitable for the practical application scenario of mechanical fault detection, which is is specifically an improvement for ResNet. The revision could be found at line 110, page 3:
However, it still lacks a solution with integrated, accurate and robust internal gear pump fault detection and RUL prediction. Referring to deep neural networks for image processing, we use the CNN for health state classification and available remaining life prediction tasks. In CNNs, LeNet[29] extracts features by using a combination of successive convolutional and pooling layers. AlexNet[30] proposed by Alex Krizhevsky et al. pioneered the use of deep neural networks to solve image problems, AlexNet has a deeper network structure compared to LeNet, using data augmentation, dropout to suppress overfitting, and ReLU activation function to reduce gradient disappearance. The AlexNet model achieves better results by constructing a multilayer network, but does not provide a direction for deep neural network design. VGG[31] constructs deep convolutional neural networks by using a series of small-sized 3x3 convolutional kernels and pooling layers, and achieves good results. InceptionNet[32] introduced the Inception block to use different sized convolutional kernels within the same layer of the network, improving model perception, using batch normalization and alleviating the gradient disappearance problem. However, with the continuous development of deep learning, the number of layers of the model is increasing and the network structure is becoming more and more complex. ResNet proposes residual learning to solve the problem that the training error increases instead of decreasing as the number of layers of the network increases, based on the problems of the above models.
- Page 7-line 221. How the convergence of the model is guaranteed in two stages should be discussed.
Response: Thank you very much for your suggestion. We illustrated the convergence of the model by analysis of the accuracy graph in the manuscript on line 401, page 11:
Proposed model with smaller was better than the model with , which means that the smaller improves the performance of ResNet model. In the CRWU dataset, the proposed method still outerperformed all the state-of-the-art methods, with improved accuracy of 0.09%-1.42%, as shown in Figure 6 and Figure 7, it can be seen that the accuracy curve increases rapidly with the increase of training times, and gradually stabilizes, the mode gradually converges, and the results of cross-validation can also show that the model is stable and the model is robust, which means that the proposed method are robust and could be extended to other fault diagnosis tasks, shown in Table 5.
- Page 10-line 311. The training hyper-parameters (e.g. batch size, etc) of the deep learning model are ignored. The division ratio of the training set and test set and the number of samples should also be reported.
Response: Thank you very much for your suggestion. The hyperparameters of the model are all tuned by manual optimisation, while the sample sizes of the data at different sample lengths are described in detail in Table 3, including the division between the training and test sets, with our test set ratio being 0.2,we described in the main text on line 378, page 11:
We tested two proposed models with and in the fault detection stage named FD-H0.1 and FD-H1.0, respectively. Six proposed models with were also tested in the RUL prediction stage with different sample lengths. The accuracy of the proposed models were compared with other deep learning models and methods from previous works. The inference time of all the proposed models were also tested to check if the proposed method could meet the needs of real-time monitoring. The parameter settings and data sampling result used in the experiments are shown in Table 3. Five-fold cross-validation was used in all experiments. The hyperparameters used in the model were manually optimised and adjusted during the experiment.
- Page 14-line 401. Since the author reported the reasoning speed, the experimental platform should be given.
Response: Thank you very much for your suggestion.We provided the description of the experimental platform for inference speed in the manuscript on line 386, page 11 and on line 455, page14:
The experiments were conducted using Python 3.6 (Python Software Foundation, Delaware, United States, http://www.python.org) and PyTorch 1.8 (Facebook AI Research, New York, United States, https://pytorch.org/) on an Nvidia RTX 2060 GPU (Nvidia Corporation, Santa Clara, California, United States).
To be more informative, we compared the sampling time and inference time of individual samples in each model. The inference speed of the models was tested in a cpu environment by reading the trained models. We used a single-core 2.5GHz CPU in this experiment. The sampling frequency of the self-collected dataset was 65536 samples per second.
- Page 12-line 355. The author gives the optimization curve, but should supplement the loss function of the model.
Response: Thank you very much for your suggestion.We have added the formula for the loss function in the manuscript on page 6, line 226:
The loss function for both stages of the model is a cross-entropy loss, and the loss function is shown in equation (12).
|
(12) |
Where denotes the number of samples.
- Page 1-line 31. Deep learning has demonstrated superior performance in many fields, including object detection, segmentation, and recognition. The following related work of deep learning for generalization and applications must be cited in Introduction, including “Improvement of generalization ability of deep CNN via implicit regularization in two-stage training process,” IEEE Access, vol. 6, pp. 15844-15869, 2018. “Compound figure separation of biomedical images with side loss.” Deep Generative Models, and Data Augmentation, Labelling, and Imperfections. Springer, Cham, 2021. 173-183. VoxelEmbed: 3D instance segmentation and tracking with voxel embedding based deep learning. International Workshop on Machine Learning in Medical Imaging. Springer, Cham, 2021: 437-446.“Pseudo RGB-D Face Recognition,” in IEEE Sensors Journal, vol. 22, no. 22, pp. 21780-21794, 15 Nov.15, 2022.
Response: Thank you very much for your suggestion. We have cited this literature in our paper.
12.Yao, Tianyuan, et al. "Compound figure separation of biomedical images with side loss." Deep Generative Models, and Data Augmentation, Labelling, and Imperfections. Springer, Cham, 2021. 173-183.
13.Zhao, Mengyang, et al. "VoxelEmbed: 3D instance segmentation and tracking with voxel embedding based deep learning." International Workshop on Machine Learning in Medical Imaging. Springer, Cham, 2021.
14.Jin, Bo, Leandro Cruz, and Nuno Gonçalves. "Pseudo RGB-D Face Recognition." IEEE Sensors Journal 22.22 (2022): 21780-21794.
21.Zheng, Qinghe, et al. "Improvement of generalization ability of deep CNN via implicit regularization in two-stage training process." IEEE Access 6 (2018): 15844-15869.
We have cited these documents in the manuscript on line 82 and line 86, page 3:
Deep learning methods have achieved great success in various fields [10, 11], include object detection, segmentation and recognition[12,13,14]. Life prediction and fault diagnosis models based on deep learning can effectively analyse and process these characteristic signals and are already a popular method for predicting the fault and RUL of machinery [15,16].
Typical deep learning methods include Deep Belief Network (DBN) [17], Autoencoder (AE) [18], Convolutional Neural Network (CNN) [19,20,21] and Recurrent Neural Network (RNN) [22,23] and others.

Reviewer 3 Report
The authors have performed an experimental study on Fault Detection and Remaining Use-2 ful Life Prediction Model of Internal Gear Pumps.
The following comments can be addressed to enhance the quality of the article.
1. Kindly abbreviate the acronyms (CEEMD, SVD, EMD etc) used in the introduction part.
2. Please check line no. 94. it is DCAE or DCSE?
3. What is the difference between the presented work and the work reported in Ref. 27?
4. The research gap and novelty of the work are not described in the article.
5. The authors selected the Robust-ResNet algorithm. Why?
6. The other algorithms are not considered. Why?
7. All the equations have to be cited properly.
8. [Error! Reference source not found] error has to be fixed with the reference manager.
9. Kindly avoid the term 'we' in the manuscript.
10. In table 4, the authors varied the Learning Rate, Epoch, and Optimizer. But these parameters are not tested for all three models. Why? The authors have to vary all the parameters for all three models.
11. Figure quality needs to be improved.
12. Presented results have to be discussed separately. A separate discussion section with the research essence has to be added.
13. Conclusion needs to be revised with research findings and outcomes.
14. Ref 36. Add the URL for the dataset.
15. References 15,16,18, and 27 are not in the correct format.

Author Response
RESPONSES TO REVIEWER3
- Kindly abbreviate the acronyms (CEEMD, SVD, EMD etc) used in the introduction part.
Response: Thank you very much for your suggestion. We have explained the abbreviations on line 64, page 2, in the manuscript.
For example, Miao et al. [5] used Complementary Ensemble Empirical Mode Decomposition (CEEMD) and Singular Value Decomposition (SVD) to decompose pressure signal, vibration signal, and flow signal to construct feature vectors and built a hydraulic pump fault diagnosis method through a TrAdaBoost migration learning algorithm. Lu et al. [6] decomposed the piston pump outlet pressure signal based on Empirical Mode Decomposition (EMD) and filtered the important Intrinsic Mode Function (IMF) components according to the correlation coefficient criterion, and calculated the characteristic energy entropy of each component as the fault feature vector. Yang et al. [7] combined Ensemble Empirical Mode Decomposition (EEMD) signal decomposition and sample entropy to extract rotating machinery fault features. Wang et al [8] used multiscale alignment entropy as a feature extraction tool for hydraulic pump and bearing fault diagnosis, respectively. Zhou et al [9] used Variational Mode Decomposition (VMD) to decompose, filter, and reconstruct the plunger pump vibration signal, and extracted fine composite multiscale fluctuating dispersion entropy for the processed signal as a feature vector.
- Please check line no. 94. it is DCAE or DCSE?
Response: Thank you very much for your suggestion. We have modified the translation of the Deep Convolutional Autoencoder and the final abbreviation is DCAE.
And in line 100,page 2:
Wang et al [27] proposed a scheme to predict the remaining life of a hydraulic pump (internal gear pump) by combining a deep convolutional autoencoder (DCAE) and a bi-directional long and short-term memory (Bi-LSTM) network.
- What is the difference between the presented work and the work reported in Ref. 27?
Response: Thank you very much for your suggestion. The cited paper "Towards robust ResNet: A small step but a giant leap" focuses on the use of a step factor h to enhance the robustness of ResNet, and our work focuses on the health status of gear pump data based on this approach The focus of our work is on the two tasks of classifying the health status of gear pump data and predicting the remaining usable life. The convolution of the model is also changed to 1D to fit the dataset and task characteristics, and the excellent effectiveness and feasibility of this approach in this domain is verified by two-stage multi-channel experiments. The generalisability of this approach in the field of fault detection and RUL is also illustrated.
- The research gap and novelty of the work are not described in the article.
Response: Thank you very much for your suggestion. The novelty of the method and its characteristics have been described in the manuscript on line 137, page 3:
The contributions made by the model proposed in this paper, as well as the novelty, are as follows.
(1)A multi-channel two-stage Robust-ResNet-based deep learning method for fault detection and RUL prediction is proposed, which does not require manual feature extraction or complex data processing.
(2)The proposed method is based on the Robust-ResNet algorithm with a small step factor , and the robustness of the proposed method is proved in experiments of fault detection and RUL prediction in two datasets. Robust-ResNet was used in mechanical fault diagnosis and RUL prediction for the first time. Its superior performance is more in line with the actual scenario of mechanical fault detection. When acquiring data in mechanical work, it is inevitable that noisy data will be obtained, while the proposed model is robust, more stable and more suitable for training data for mechanical fault detection.
(3)The proposed method used multi-channel signals in RUL prediction, including pressure pulsation and vibration signals. The fusion of multi-channel signals significantly improved the prediction performance of the method.
(4)The model achieves high accuracy in its performance on both tasks, reaching 99.96% accuracy in the fault detection task and 99.53% accuracy in the RUL prediction task in the self-collected internal gear pump dataset. It could also used in the fault detection for rolling bearing, which achieved 99.94% accuracy in CWRU[34] dataset, which outperformed other status-of-the-art methods. The superior performance of the method on both datasets demonstrates its suitability for multiple application scenarios and can be generalised for practical use.
- The authors selected the Robust-ResNet algorithm. Why?
Response: Thank you very much for your suggestion. Firstly ResNet is mainly designed to solve the problem that the error of deep neural networks rises rather than falls as the number of layers of the network increases. Robust-ResNet, on the other hand, uses a step factor h to improve the robustness of ResNet on top of this. In practical gear pump health state classification and RUL tasks, the acquired data will inevitably contain noisy data, which will have an impact on model training, while the robustness of Robust-ResNet enables it to avoid the impact of noisy data on training, showing that that the method is more suitable for practical application environments.
- The other algorithms are not considered. Why?
Response: Thank you very much for your suggestion. We also compared a number of other methods, including those implemented in the previous sensors paper, so we did consider other algorithms, but found that this method worked best.
- All the equations have to be cited properly.
Response: Thank you very much for your suggestion. We have modified the citation format of the formulae in the manuscript, modified the formulae to indicate that the opening letters are capitalised in English, modified the wrap-around type of the formulae table to none and the left indent to 4.6 cm, and modified the formulae to have no indent at the bottom first line.
For example, the Equation on line 203, page 5 of the manuscript:
Each residual block can be expressed as Equation (1).
|
|
(1) |
where and denote the input and output of the one residual block , respectively. is the residual function, which weighted the input . is the Relu activation function. Based on the above equation, the learning features from shallow block to deep block could be found in Equation (2):
|
(2) |
In the above example the first paragraph of text is indented by 0.75cm, the following formula corresponds to the text, the wrap type is none and the indent is 0cm. the first paragraph of text below the formula is not indented.
- [Error! Reference source not found] error has to be fixed with the reference manager.
Response: Thank you very much for your suggestion. The incorrect citation formatting has been corrected on line 280, page 7 of the manuscript. The revised article reads as follows.
The signals commonly used to monitor and reflect the status and performance of hydraulic pumps include vibration signals [36,37], flow signals [38], pressure signals [39,40], and pressure pulsation signals [41], among which vibration signal analysis is most widely used. According to statistics, vibration signal analysis accounts for about 70% of the existing mechanical fault diagnosis system [42].
- Kindly avoid the term 'we' in the manuscript.
Response: Thank you very much for your suggestion. We have revised some of the statements in the manuscript.
The rest of this paper is arranged as follows. Section 2 describes the details of the method proposed by this work. The experimental setup and results are described and discussed in Section 3. Finally, conclusion is given in section 4.
For the RUL prediction stage in the figure, a sample length of 1024 is used as an example. In practice, different sample lengths are compared in the actual experiment.
- In table 4, the authors varied the Learning Rate, Epoch, and Optimizer. But these parameters are not tested for all three models. Why? The authors have to vary all the parameters for all three models.
Response: Thank you very much for your suggestion. The parameters used in our experiments were chosen after several experiments to be optimal for the current model.
- Figure quality needs to be improved.
Response: Thank you very much for your suggestion. The resolution of the image has been increased in the manuscript.
- Presented results have to be discussed separately. A separate discussion section with the research essence has to be added.
Response: Thank you very much for your suggestion.A discussion section has been added to line 453 on page 14 of the manuscript.
3.5 Discussion
CNN is mainly used to train the gear pump data, to realize the gear pump health status classification and RUL task. In convolutional neural networks such as AlexNet, VGG and InceptionNet, the more layers there are, the richer features of different levels can be extracted. The deeper the network is, the more abstract the extracted features are and the more semantic information they have. However, with the increase in the number of network layers, the model structure gradually becomes complex, but the network is faced with the problem of degradation, and the model error increases. Based on this problem, ResNet proposed residual learning, which does not directly fit the output value but learns an identity mapping. However, in the actual working environment of gear pump health status classification, the acquisition of data will inevitably obtain noise data, which will cause adverse effects on the model and affect the final model performance. Such errors may cause significant losses in the actual environment. Therefore, based on ResNet, step factor is used in this paper to improve the robustness of ResNet and reduce the impact of noise data on the model. More in line with the actual application situation. In the health status classification and RUL tasks, according to the accuracy curve, it is obvious that the proposed method can converge quickly to reach a stable state, and has very high accuracy on both data sets, which also proves that our method can be extended to different application scenarios, not limited to the health status management of gear pumps.
- Conclusion needs to be revised with research findings and outcomes.
Response: Thank you very much for your suggestion.The conclusion has been revised in the manuscript on line 488, page 15:
In the task of health status classification, the model is compared with the network structure of CNN, CNN+Attention, etc., and it can be seen that this method is superior to other methods, which verifies the superiority of this method in the task of gear pump health status management. Compared with other models, it can better avoid the influence caused by noisy data. The accuracy of this model for the multi-channel signal is improved by 6.2% ~ 15.49%, which is significantly better than that of other methods, and the accuracy of this method is improved by 0.7% ~ 23.93%. This improvement means that the feature extraction capability of this method is significantly superior to other deep learning methods. According to the experimental results, the proposed method can not only reduce the influence of noise data but also extract multi-channel signals with high feature extraction ability. These two characteristics make the method not only suitable for the gear pump data health state management but also can be extended to other application scenarios.
- Ref 36. Add the URL for the dataset.
Response: Thank you very much for your suggestion. URL has been added to the reference.
- Loparo, K.A. Bearings Vibration Data Set; The Case Western Reserve University Bearing Data Center. Available online: http: //www.eecs.cwru.edu/laboratory/bearing/download.htm (accessed on 1 May 2022).
- References 15,16,18, and 27 are not in the correct format.
Response: Thank you very much for your suggestion. We have corrected the incorrectly formatted references.With the inclusion of citations, the numbering of documents in the manuscript has changed.
The incorrect literature citation is as follows.
15.TSCHANNEN M, BACHEM O, LUCIC M. Recent advances in autoencoder-based representation learning [J]. arXiv preprint arXiv:181205069, 2018.
16.O'SHEA K, NASH R. An introduction to convolutional neural networks [J]. arXiv preprint arXiv:151108458, 2015.
18.ZAREMBA W, SUTSKEVER I, VINYALS O. Recurrent neural network regularization [J]. arXiv preprint arXiv:14092329, 2014.
27.Zhang, J., Han, B., Wynter, L., Low, K. H., & Kankanhalli, M. (2019). Towards robust ResNet: A small step but a giant leap. arXiv preprint arXiv:1902.10887.
The revised literature citation is as follows.
18.Tschannen, Michael, Olivier Bachem, and Mario Lucic. "Recent Advances in Autoencoder-Based Representation Learning."
19.O’Shea, Keiron, and Ryan Nash. "An Introduction to Convolutional Neural Networks."
22.Zaremba, Wojciech, Ilya Sutskever, and Oriol Vinyals. "Recurrent Neural Network Regularization." (2014).
33.Zhang, Jingfeng, Bo Han, Laura Wynter, Bryan Kian Hsiang Low, and Mohan S. Kankanhalli. "Towards Robust ResNet: A Small Step but a Giant Leap." In IJCAI. 2019.

Round 2
Reviewer 1 Report
my comments have been addressed. it can be accepted
Reviewer 2 Report
Accept.
Reviewer 3 Report
Greetings to the authors.